# Mediation Role of Behavioral Decision-Making Between Self-Efficacy and Self-Management Among Elderly Stroke Survivors in China: Cross-Sectional Study

**DOI:** 10.3390/healthcare13070704

**Published:** 2025-03-23

**Authors:** Xiaoxuan Wang, Hu Jiang, Zhixin Zhao, Noubessi Tchekwagep Kevine, Baoxia An, Zhiguang Ping, Beilei Lin, Zhenxiang Zhang

**Affiliations:** 1Nursing and Health School, Zhengzhou University, Zhengzhou 450001, China; wangxiaoxuan222666@163.com (X.W.); kevine3noubessi@gmail.com (N.T.K.); 2Henan Huaxian People Hospital, Anyang 456400, China; 3College of Public Health, Zhengzhou University, Zhengzhou 450001, China

**Keywords:** stroke, self-efficacy, behavioral decision-making, self-management

## Abstract

**Background:** Identifying the factors that impact self-management is crucial, as elderly stroke survivors frequently face challenges in self-management. Self-efficacy and behavioral decision-making are reported as influencing factors of self-management, but their relationship within the elderly population remains unconfirmed. This study aimed to explore whether self-efficacy impacts self-management through the mediating role of behavioral decision-making among elderly stroke survivors. **Methods:** A cross-sectional design and convenience sampling method were used in this study. A total of 291 elderly stroke survivors were recruited from a tertiary hospital in Henan Province, China, between March and July of 2024. Questionnaires were distributed to collect sociodemographic, self-efficacy, behavioral decision-making, and self-management data. A path analysis and correlation analysis were used to analyze the data. This study adhered to the Strengthening the Reporting of Observational Studies in Epidemiology (STROBE) guidelines. **Results:** Elderly stroke survivors reported having a moderate level of self-management. There was a positive correlation between self-efficacy, behavioral decision-making, and self-management (all *p* < 0.01). The mediation model indicated that behavioral decision-making mediated the association of self-efficacy and self-management in the regression model (95% *CI* 0.03 to 0.14), and the effect value was 0.08. It was also confirmed that behavioral decision-making mediated the impact of self-efficacy and self-management, accounting for 25.81% of the total effect. **Conclusion:** Self-efficacy is not solely a key factor influencing self-management in elderly stroke survivors, but it also improves their self-management behaviors by facilitating behavioral decision-making. As a result, healthcare professionals should consider self-efficacy and behavioral decision-making as crucial elements for assessing elderly stroke survivors during discharge and follow-up.

## 1. Introduction

The latest Global Burden of Disease (GBD) report indicates that the number of global stroke cases has surpassed a notably high 93 million [1]. China has the largest number of stroke survivors in this statistic, and the number of stroke survivors exceeds 28 million [1]. Among these stroke survivors, a substantial proportion, accounting for 50.81%, is over 60 years old [2]. In other words, the proportion of elderly stroke survivors in China exceeds more than 14 million. Furthermore, with the continuous acceleration of the aging population in the past decade in China, the number of elderly stroke survivors has been increasing [3]. Therefore, it is pertinent for researchers to focus on elderly stroke survivors.

The advancement in medical care and healthcare has led to the early detection and rapid treatment of stroke, thus reducing morbidity and mortality rates among stroke survivors. However, most stroke survivors, especially the elderly with multiple chronic diseases, cognitive impairment, and poor daily living activity, often face significant challenges in disease management and rehabilitation after returning home [4,5]. In addition, advanced age has also been proven to be a risk factor for post-stroke depression [6] and post-stroke fatigue [7]. Therefore, we may deduce that disease management for elderly stroke survivors is quite challenging.

Self-management has been identified as a crucial model for elderly care [8]. The components of self-management post-stroke have been delineated as follows: disease management, safe medication practices, diet management, daily living management, rehabilitation exercises, and so on [9]. However, a previous study has found that elderly stroke survivors have a lower level of self-management [10]. Exploring the factors influencing self-management among elderly stroke survivors is essential for designing interventions to improve their self-management.

Self-efficacy refers to an individual’s judgment of their own ability to organize and execute the action processes required to achieve specific behavioral goals [11]. Stroke survivors with higher self-efficacy tend to have stronger motivation and engage in self-management more effectively [12]. Given that elderly stroke survivors often experience diminished physical function and lower self-efficacy compared with younger patients [13], the relationship between self-efficacy and self-management is particularly critical.

Additionally, behavioral decision-making is another important factor influencing stroke self-management. The process of behavioral change relies on scientific decision-making [14]. Behavioral decision-making is an interdisciplinary concept that originated from the behavioral decision theory established by Edwards in the 1950s [15]. Our research team previously explored the behavioral decision-making of stroke survivors [16], defining stroke behavioral decision-making as the process by which stroke survivors, under the influence of social, economic, and cultural environments, comprehensively weigh their own needs, expectations, and environmental factors. Guo et al. [17] indicated that elderly stroke survivors rarely make proactive rehabilitation decisions in the early stages of recovery.

A meta-analysis has found that self-efficacy is an important variable in health decision-making [18]. Our research group previously integrated a literature review, qualitative research, and theoretical analysis to preliminarily construct a situational theoretical Recurrence Risk Perception and Behavioral Decision Model in stroke survivors [16], which was in alignment with a subsequent study [19]. This model can guide how behavioral surveys are conducted among stroke survivors. It found that stroke survivors undergo a series of internal decision-making processes when they decide to adopt healthy behaviors. Moreover, the process of behavioral decision-making is affected by self-efficacy [16]. Therefore, we speculate that the self-efficacy of elderly stroke survivors will influence self-management by affecting behavioral decision-making. Previous studies have only explored the relationships between two variables [12,18,20]. However, the relationship between these three variables is not yet clear, especially in the population of elderly stroke survivors. Therefore, our study aimed to explore the relationships between self-efficacy, behavioral decision-making, and self-management among elderly stroke survivors, guided by the stroke behavioral decision-making model. In addition, this study will provide new perspectives for developing behavioral management measures for elderly stroke survivors. The research hypotheses are as follows, as shown in Figure 1:

**Hypothesis** **1.***Self-efficacy is positively correlated with self-management*.

**Hypothesis** **2.***Self-efficacy is positively associated with decision-making*.

**Hypothesis** **3.***Behavioral decision-making positively and significantly relates to self-management*.

**Hypothesis** **4.***Behavioral decision-making partially mediates the relationships between self-efficacy and self-management*.

## 2. Materials and Methods

### 2.1. Design

This was a cross-sectional study, and convenience sampling was used. We followed the guidelines for Strengthening the Reporting of Observational Studies in Epidemiology (STROBE, see Appendix A).

### 2.2. Sample

This study was conducted among hospitalized stroke survivors at a tertiary hospital in Luoyang, Henan Province, China. Eligible elderly stroke survivors who complied with this study’s inclusion criteria were invited to participate in this voluntary study. Inclusion criteria for elderly stroke survivors comprised the following: (a) over 60 years old; (b) diagnosed with stroke by MRI or CT [21]; (c) in the stages of recovery from stroke; and (d) exhibited normal language abilities (Token Test score ≥ 17) and cognitive function (Mini-Mental State Examination, MMSE score ≥ 27). Exclusion criteria for elderly stroke survivors comprised the following: (a) having severe cardiac, liver, or renal dysfunction or other malignant tumors or (b) having a history of serious mental illness or a family history of serious mental illness.

According to Liu [22], the sample size was 5–10 times greater than the number of independent variables. There were 26 variables included in this study, which were as follows: 12 general information questions, 7 dimensions of the stroke self-management scale, 2 dimensions of the stroke self-efficacy questionnaire, and 4 dimensions of the behavioral decision-making scale for stroke survivors. The modified Rankin scale was also used. Considering that 10% of questionnaires were invalid, the minimum sample size of this study was 288 stroke survivors; hence, a sample size of 291 met the requirement for testing the hypothesis models.

### 2.3. Measurements

#### 2.3.1. General Information Questionnaire

A demographic and disease-specific data questionnaire was designed and used to collect information on gender, age, marriage, residential area, working state, educational level, number of strokes, duration of stroke, family history of stroke, type of stroke, number of chronic diseases, and activity of daily living (ADL).

#### 2.3.2. The Modified Rankin Scale (MRS)

The modified Rankin scale was developed by Rankin [23] in 1957 for assessing the neurological recovery of stroke survivors. It consists of six levels with a total score ranging from 0 to 5, where a score of 0 indicates no symptoms and no assistance required, while a score of 5 signifies severe disability and complete dependence on others for assistance. A score less than 3 indicates a favorable prognosis, and a score of 3 or above suggests a poor prognosis [24]. The Cronbach’s alpha value of this scale is 0.773.

#### 2.3.3. The Stroke Self-Management Scale (SSMS)

The stroke self-management scale was developed by Wang et al. [25] in 2013. It includes 7 dimensions and 50 items, which are disease management (11 items), safe medication management (5 items), dietary management (8 items), daily living management (8 items), emotion management (5 items), social functioning and interpersonal management (6 items), and rehabilitation exercise management (7 items). It is a 5-point Likert scale. The total score ranges from 50 to 250 points. Higher scale scores mean higher levels of self-management. The Cronbach’s alpha value of the SSMS is 0.874.

#### 2.3.4. The Stroke Self-Efficacy Questionnaire (SSEQ)

The stroke self-efficacy questionnaire was developed by Jones et al. [26] in 2008. This questionnaire comprises 2 dimensions: movement (8 items) and self-management (5 items). Each item is rated on an 11-point Likert scale ranging from 0 (‘little confidence’) to 10 (‘strong confidence’). Li [27] revised the SSEQ into a Chinese version and deleted 2 items in 2015. The total score ranges from 0 to 110 points. Higher scale scores mean higher levels of self-efficacy. The Cronbach’s alpha value of the SSEQ is 0.969.

#### 2.3.5. The Behavioral Decision-Making Scale for Stroke Patients

The behavioral decision-making scale for stroke patients was developed by Lin et al. [28] in 2022. It is divided into four dimensions and consists of 29 items: behavioral change motivation (10 items), behavioral change intention (9 items), decision-making factors (5 items), and decisional balance (5 items). It is a 5-point Likert scale ranging from 1 (‘little agreement’) to 5 (‘strong agreement’). The total score ranges from 30 to 150 points. The higher the score, the higher the level of behavioral decision-making in stroke survivors, and the easier it is to trigger healthy behavioral decisions, resulting in healthy behaviors. The Cronbach’s alpha value of this scale is 0.934.

### 2.4. Data Collection

Data collection spanned from 10 March 2024 to 15 July 2024. This research involved two trained assistants who were responsible for enrolling stroke survivors for in-person interviews. Patient identification was facilitated through medical records and hospital databases, with confirmation by the principal investigator. For this pre-testing phase, 5 stroke survivors who had only completed elementary school were selected. The assistants meticulously checked the questionnaires for any ambiguities or points needing clarification. Their insights led to modifications to ensure clear understanding and accurate responses to the survey questions in the main study. Prospective participants were briefed on this study’s objectives and provided with informed consent before participation. Upon granting written approval of the survey, 300 individuals received a set of questionnaires to complete.

### 2.5. Ethical Considerations

The experimental protocol in this study was approved by the Zhengzhou University ethics committee in China (approval number: ZZURIB2021-115) in 2021, and all methods were conducted following relevant guidelines and regulations. All participants gave written informed consent.

### 2.6. Data Analysis

A statistical analysis was conducted using SPSS version 26.0, focusing on descriptive statistics. Samples with missing data exceeding 10% were discarded. Upon conducting the Kolmogorov–Smirnov test, it was determined that the scores of self-management in this research did not adhere to a normal distribution. Consequently, these non-normally distributed figures were characterized using the median and the interquartile range. For categorical data, representation was conducted through counts and their corresponding percentages. The Mann–Whitney U test and the Kruskal–Wallis test were used to assess differences in self-management across demographic characteristics. Spearman correlation was used to examine the associations between self-efficacy, behavioral decision-making, and self-management. Model 4 in the SPSS 26.0 macros program PROCESS compiled by Hayes [29] was used to construct the mediation model with 5000 bootstrap samples.

## 3. Results

### 3.1. Common Method Bias

The results showed that 16 factors with an eigenvalue greater than 1 were co-precipitated, and the variance explained by the first factor was 24.91%, which was less than the critical standard of 40%, indicating that the common method deviation of this study was not significant.

### 3.2. Demographic Characteristics

This study recruited 300 stroke survivors, and nine questionnaires were excluded due to non-response to some questions. A total of 291 elderly stroke survivors were included, resulting in a final valid return rate of 97%. More than 67.01% of elderly stroke survivors were men. Only 58 (11.2%) survivors in the sample had a university degree or more. In addition, more than half (59.45%) of elderly stroke survivors had at least one chronic disease comorbidity. Comparative analyses of self-management based on demographic characteristics showed no statistically significant differences in self-management scores with respect to marriage, educational level, duration of stroke, family history of stroke, mRS, and ADL. Furthermore, gender, residential area, working state, the number of strokes, the type of stroke, and the number of chronic diseases were found to have statistically significant differences in terms of self-management scores. A comparison of the scales’ scores of survivors who had a stroke with different characteristics is shown in Table 1.

### 3.3. Correlations Among the Main Variables

Spearman’s correlation analysis was used to investigate the correlations among the three variables of self-efficiency, behavioral decision-making, and self-management. The results showed that self-efficacy was significantly positively correlated with behavioral decision-making (*r* = 0.355, *p* < 0.01) and self-management (*r* = 0.408, *p* < 0.01). Furthermore, behavioral decision-making was significantly positively correlated with self-management (*r* = 0.366, *p* < 0.01).

### 3.4. Mediation Model

According to the procedure and steps of mediating effect testing by Wen et al. [30], firstly, we examined the predictive effects of self-efficacy on self-management. Then, we used the bootstrap method (with 5000 resamples) to test the mediating role of behavioral decision-making. We controlled for some general demographic factors (such as gender, residential area, working state, number of strokes, type of stroke, and number of chronic diseases). Self-efficacy was positively related to self-management (*β* = 0.31, *t* = 6.50, *p* < 0.001). After adding a mediating variable, self-efficacy (*β* = 0.23, *t* = 4.75, *p* < 0.001) and behavioral decision-making (*β* = 0.23, *t* = 4.80, *p* < 0.001) were positively correlated with self-management. The results are shown in Table 2. Figure 2 presents the influencing paths of the mediation model.

To ensure the accuracy of the test, the 95% *CI* of the mediating effects of behavioral decision-making was 0.03 to 0.14, which does not contain 0, indicating that the mediating effects of behavioral decision-making between self-efficacy and self-management were established, and the effects accounted for 25.81%.

## 4. Discussion

This study reported on the influencing factors of self-management among elderly stroke survivors and the relationships between these factors. To the best of our knowledge, this is the first study to examine the relationships between self-efficacy, behavioral decision-making, and self-management. Furthermore, our study expanded the application scope of the behavioral decision-making model for stroke and validated its applicability within the elderly stroke survivor population.

Also, the findings of our study indicated that the self-management levels of elderly stroke survivors were moderately high, which was consistent with previous research [31]. Among them, the daily living management scores for stroke survivors were relatively high, while the scores for rehabilitation exercise management and disease management were relatively low. This may be because the vast majority (96.91%) of stroke survivors in our study had relatively high ADL scores, allowing them to achieve a high level of self-care in their daily lives. As a result, their daily living management was not significantly affected by stroke. On the contrary, disease management requires elderly stroke survivors to quickly adapt to the role of a patient and be able to persist in disease monitoring and management. However, elderly stroke survivors often experience a decline in memory and cognitive functions [32,33] due to factors like age, which may lead to forgetfulness or the insufficient mastery of disease monitoring skills in tasks such as blood pressure and blood sugar monitoring. In addition, a previous longitudinal study in China found that physical activity emerged as a significant predictor of decreased daily living activities among older adults [34], which was consistent with our study. However, elderly stroke survivors are a high-risk population for frailty [35], falls [36], and sarcopenia [37]. Risk factors such as frailty and falls can severely impact the ADL of elderly stroke survivors, thereby reducing their self-efficacy and self-management. Therefore, health professionals should implement the assessment of these weakness and risk factors in elderly stroke survivors. They should also provide these individuals with proper guidance and health education to help them prevent incidents such as falls while participating in exercise and disease management.

Our study indicated that self-efficacy can positively affect self-management among elderly stroke survivors. This result is consistent with previous research results [12,38,39]. The reasons for this may be that elderly stroke survivors with higher self-efficacy have intrinsic motivation, thus having more confidence to engage in self-management behaviors [12]. According to the Health Action Process Approach (HAPA) theory [40], self-efficacy is a crucial aspect in all stages of behavior changes, whether in the context of the emergence of behavioral intentions, the execution of behavior, or overcoming difficulties in the process of behavior. Elderly stroke survivors often have poorer physical function and less confidence in stroke management and rehabilitation. In other words, self-efficacy plays a more significant role in the rehabilitation process among elderly stroke survivors. Consequently, healthcare professionals should incorporate self-efficacy into a multifaceted evaluation framework for elderly stroke survivors to better grasp their disease conditions.

Moreover, our study found that the relationship between self-efficacy and self-management was also influenced by a mediator, namely behavioral decision-making. In other words, elderly stroke survivors with higher self-efficacy were shown to have a higher level of behavioral decision-making, which improved their engagement in self-management more effectively. This might be because elderly stroke survivors with higher self-efficacy have more control over their condition [12], which in turn allows them to have stronger behavioral motivation and intention when making decisions regarding self-management behaviors. Specifically, elderly stroke survivors are better able to juggle their own needs and make decisions that are in their best interest, thereby adopting self-management behaviors more effectively. The role of behavioral decision-making among stroke survivors has been confirmed in previous studies [17,41]. Guo et al. [17] found that the difficulty of behavioral decision-making is a barrier to prompting stroke survivors to engage in self-management behaviors.

However, behavioral decision-making is an extremely complex process [42]. After being discharged and returning home, elderly stroke survivors are often overprotected by caregivers [43] due to their advanced age. This reduces the autonomy and initiative of elderly stroke survivors in making their own decisions. Therefore, autonomy is paramount in behavioral decision-making among elderly stroke survivors and in turn crucial for their behavioral changes. Considering the decline in physical and cognitive abilities of elderly stroke survivors, to ensure efficient decision-making, the assistance of healthcare professionals and family members should be utilized while protecting the decision-making autonomy of elderly stroke survivors. Shared decision-making (SDM) is a patient-centered model in which doctors and patients share information, discuss options, and make decisions together based on mutual respect and equality [44]. This approach aims to better meet patients’ needs and improve their treatment experience. American cardiovascular societies have all endorsed shared decision-making [45]. Healthcare professionals and family members should fully take into consideration the preferential needs of elderly stroke survivors and collaborate synergistically with them to make the most suitable health decisions [46]. Patient Decision Aids (PDAs) [47] are evidence-based tools that provide patients with decision-relevant information, assist them in weighing the pros and cons, and help them in making informed decisions. Healthcare professionals can design various decision aids, such as question prompt lists [48], to help elderly stroke survivors make careful decisions and reduce decisional conflict. However, it should be noted that the surveyed carried out by He et al. [20] included 229 stroke survivors and found that elderly stroke survivors have a higher level of behavioral decision-making compared to middle-aged and younger stroke survivors. Although this may not directly align with our conventional understanding and even though the sample size of this study was not large, this is indeed an objective result. Future research could consider conducting multicenter, large-sample surveys to assess the behavioral decision-making levels of elderly stroke survivors.

### 4.1. Clinical Implications

This study offers a new perspective on stroke behavior management. Firstly, the role of self-efficacy in promoting self-management deserves attention. For elderly stroke survivors, the establishment of self-efficacy may be more challenging. According to self-efficacy theory [49], successful experiences are one of the most important ways to gain self-efficacy. Therefore, healthcare professionals may consider setting gradual rehabilitation goals tailored to the conditions of elderly stroke survivors, allowing them to build confidence through successful management. Additionally, the significant role of behavioral decision-making among elderly stroke survivors was proven in our study. Hence, healthcare professionals should convey to the families of elderly stroke survivors the importance of patients’ independent ability to make decisions with the help of family members [50].

### 4.2. Limitations

This study has certain limitations. Firstly, it is a cross-sectional study, so we cannot establish causal relationships between variables. Additionally, the elderly stroke survivors included in this study had a relatively high ADL level. This may, to some extent, contribute to Type I errors. Typically, most elderly stroke survivors experience significant functional impairments following a stroke, resulting in decreased ADL levels and compromised self-management abilities. Therefore, the generalizability of our study’s findings is limited and cannot be extended to elderly stroke survivors with lower ADL levels. Moreover, although this study adequately considered potential influencing factors, there are some unique factors affecting the self-management of older adults that were not fully considered, such as falls and frailty related to the decline in physical function in old age. Future research could focus on the impact of these factors on the self-efficacy and self-management of elderly stroke survivors. Lastly, this study concentrated on how the inherent characteristics of elderly stroke survivors affect their self-management; thus, all the factors examined pertain to the survivors themselves. Individual behavior is also influenced by family and societal factors [31]. Consequently, future research will incorporate family and social elements pertaining to elderly stroke survivors to investigate the multifaceted influences on behavioral management.

## 5. Conclusions

This study explored the impact of self-efficacy, behavioral decision-making, and self-management on stroke survivors. In summary, self-efficacy and behavioral decision-making are significantly related to self-management. Behavioral decision-making played a mediation role in the relationship between self-efficacy and self-management among elderly stroke survivors. Furthermore, this study indicated that elderly stroke survivors were more likely to adopt behavioral decision-making to improve self-management when they have higher self-efficacy. These findings suggest a new perspective for healthcare professionals in elderly stroke behavior management. While enhancing the self-efficacy levels of elderly stroke survivors, it is also necessary to consider how to assist them in making behavioral decisions that align with their health rights and interests.

## Figures and Tables

**Figure 1 healthcare-13-00704-f001:**
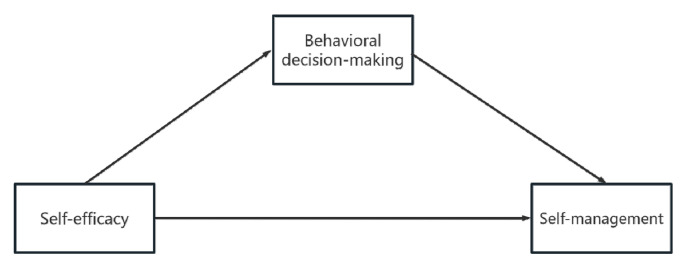
The hypothesis model of self-efficacy, behavioral decision-making, and self-management in elderly stroke survivors.

**Figure 2 healthcare-13-00704-f002:**
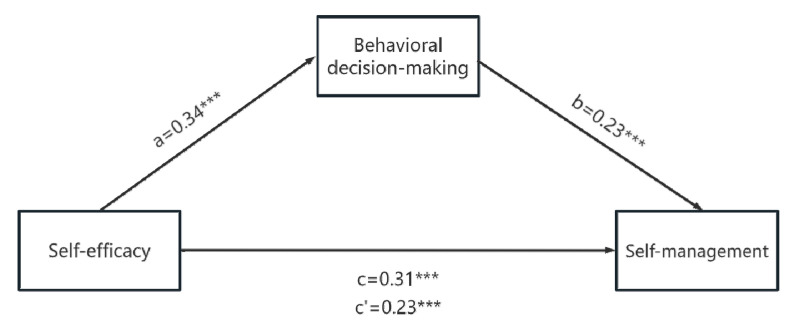
Mediating effects of behavioral decision-making on relationship between self-efficacy and self-management (*** *p* < 0.001). Numbers associated with a, b, c, and c′ are unstandardized regression coefficients. c: total effects of self-efficacy on self-management; c′: direct effects of self-efficacy on self-management.

**Table 1 healthcare-13-00704-t001:** Differences in terms of self-management in demographic factors of elderly stroke survivors (*n* = 291).

Variables	Category	*n* (%)	Self-Management [M(P25, P75)]	Z/H
Gender	Men	195 (67.01)	174 (157, 193)	−2.504 *
	Women	96 (32.99)	187.5 (161.25, 209.5)	
Marriage	Married	259 (89.00)	180 (157, 198)	−0.006
	Divorced	32 (11.00)	174 (158, 203)	
Residential area	Rural	108 (37.11)	191 (165, 209.5)	−4.288 ***
	Urban	183 (62.89)	170 (156, 191.25)	
Working state	Unemployed	105 (36.08)	184 (163, 201.75)	12.514 **
	Pensioner	84 (28.87)	189.5 (149.25, 211.75)	
	Working	102 (35.05)	164.5 (157, 190.25)	
Educational level	Primary or below	112 (38.49)	171 (156, 191)	−0.010
	Junior high school	78 (26.80)	177 (156, 201)	
	High school	79 (27.15)	186 (161, 198)	
	University or above	22 (7.56)	208 (182.25, 225.25)	
Number of strokes	1	156 (53.60)	188 (162, 200)	9.117 *
	2	95 (32.65)	165 (156, 191)	
	3 or more	40 (13.75)	169 (144, 192)	
Duration of stroke	<3 months	55 (18.90)	188 (159, 208)	3.988
	3 months~	131 (45.02)	176 (160, 193)	
	1 year~	44 (15.12)	172.5 (146.75, 199.75)	
	3 years or more	61 (20.96)	182 (156.25, 212)	
Family history of stroke	Yes	58 (19.93)	181 (146.75, 205)	−0.344
	No	233 (80.07)	177 (159, 197.75)	
Type of stroke	Ischemic	239 (82.13)	177 (157, 196)	−2.043 *
	Hemorrhagic	52 (17.87)	189 (162, 208)	
mRS (scores)	<3	243 (83.51)	184 (158, 198)	−1.708
	≥3	48 (16.49)	172 (153.25, 185)	
Number of chronic diseases	1	16 (5.50)	191 (159.5, 193.75)	15.343 ***
	2	173 (59.45)	172 (156, 192.5)	
	3	56 (19.24)	175 (156.25, 201.5)	
	4 or more	46 (15.81)	199 (175.5, 212)	
ADL	40~	9 (3.09)	176 (151.5, 207)	−0.010
	60~	282 (96.91)	178.5 (158, 198)	

Note: * *p* < 0.05, ** *p* < 0.01, *** *p* < 0.001.

**Table 2 healthcare-13-00704-t002:** Testing the mediation effects of behavioral decision-making in terms of the relationship between self-efficiency and self-management.

Regression Equation		Global Fit Index			Significance of Regression Coefficient	
Outcome variable	Predictor variable	R	R^2^	F	B (95% CI)	t
Self-management	Gender	0.51	0.26	14.21	0.16 (−0.05, 0.37)	1.51
	Residential area				0.34 (0.14, 0.54)	3.33 **
	Working state				−0.17 (−0.29, −0.05)	−2.88 **
	Number of strokes				−0.14 (−0.28, −0.01)	−2.02 *
	Type of stroke				0.20 (−0.05, 0.46)	1.58
	Number of chronic diseases				0.18 (0.06, 0.30)	2.92 **
	Self-efficacy				0.31 (0.22, 0.41)	6.50 ***
Behavioral decision-making	Gender	0.41	0.17	14.98	0.28 (0.03, 0.53)	2.19 *
	Residential area				0.36 (0.12, 0.60)	2.94 **
	Working state				−0.01 (−0.15, 0.13)	−0.16
	Number of strokes				0.18 (0.01, 0.35)	2.12 *
	Type of stroke				0.22 (−0.09, 0.52)	1.41
	Number of chronic diseases				−0.13 (−0.27, 0.02)	−1.73
	Self-efficacy				0.34 (0.23, 0.46)	5.98 ***
Self-management	Gender	0.56	0.32	16.28	0.10 (−0.11, 0.30)	0.94
	Residential area				0.26 (0.06, 0.46)	2.58 *
	Working state				−0.17 (−0.28, −0.06)	−2.94 **
	Number of strokes				−0.19 (−0.32, −0.05)	−2.68 **
	Type of stroke				0.15 (−0.09, 0.40)	1.23
	Number of chronic diseases				0.21 (0.09, 0.33)	3.51 **
	Self-efficacy				0.23 (0.14, 0.33)	4.75 ***
	Behavioral decision-making				0.23 (0.14, 0.33)	4.80 ***

Note: * *p* < 0.05, ** *p* < 0.01, *** *p* < 0.001. Gender, residential area, working state, number of strokes, type of stroke.

## Data Availability

The data supporting the findings of this study are available on request from the corresponding author. The data are not publicly available due to privacy or ethical restrictions.

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
