# Peer review of "Mediation Role of Behavioral Decision-Making Between Self-Efficacy and Self-Management Among Elderly Stroke Survivors in China: Cross-Sectional Study"

_healthcare, 2025, doi:10.3390/healthcare13070704_

Round 1
Reviewer 1 Report
Comments and Suggestions for Authors
The manuscript examines whether self-efficacy influences self-management in elderly stroke survivors through the mediating role of behavioral decision-making. The study addresses an important clinical issue by focusing on a vulnerable population and exploring factors that could inform more effective post-stroke care strategies. I have only a few recommendations to improve clarity for readers.
Introduction
- Terms such as “astonishing” or “a huge amount” might be replaced with more measured language (e.g., “notably high” or “a substantial proportion”).
-When introducing self-management, the text states multiple times that it is a key model for chronic disease care. Consolidate these points into one clear, concise paragraph.’
- The description of self-efficacy repeats its importance to self-management in several sentences. Consider integrating these ideas to state once clearly that “Higher self-efficacy has been linked to improved motivation and more effective self-management, a relationship that is particularly critical given that elderly stroke survivors often experience diminished physical function and lower self-efficacy compared with younger patients.”
Discussion
- There is some variation in terminology (e.g., “behavior decision-making,” “self-decision-making,” and “shared decision-making”). Consider standardizing the terms where possible or clearly defining each when they are first mentioned to avoid potential confusion.
- Minor errors (e.g., “our stdy” instead of “our study”) should be corrected. Additionally, consider replacing informal transitions such as “Besides” with more formal connectors like “Moreover” or “Furthermore.”
Reviewer 2 Report
Comments and Suggestions for Authors
The topic of your article on “Mediation role of behavioral decision-making between self-efficacy and self-management among elderly stroke survivors in China: a cross-sectional study” is interesting and has the potential to bring insight into better adaptations to environment and activities of daily living of post-stroke survivors, as well as the tailoring of the rehabilitation programs for the elders. However, there are some aspects that should be addressed:
- Please pay attention to typo: “Besides, our stdy found that the relationship between self-efficacy and self-management is also influenced by the mediator, namely behavior decision-making.”
- I suggest using past tense not present in the following phrase and throughout the text where you describe the population you included in the study: “Additionally, the elderly stroke survivors included in this study have a relatively high level of daily living activities.”
- As you already stated in your manuscript, the elderly stroke survivors included in the study had a relatively high level of daily life activities, which can lead to a type I error. Post-stroke, ADLs and IADLs are highly affected, thus the impact on self-efficacy and lower quality of life (QoL). I suggest to describe this more comprehensively in the manuscript.
- Your focus was on the elders, why did you not take into consideration factors such as frailty and falls? Do you have such information gathered? If so, I recommend to add it in the manuscript for a more comprehensive and global approach of stroke in the elders. However, I suggest either way to add some information regarding the risk of falls and frailty in the introduction and/or discussion, since these characteristics can impact self-esteem, self-efficacy, self-management and lead to more complications, therefore, decreasing ADLs, IADLs and QoL, as well as leading to higher rates of depression.
Good luck!
Comments on the Quality of English LanguageNone
Reviewer 3 Report
Comments and Suggestions for Authors
The authors of the article: “Mediation role of behavioral decision-making between self-efficacy and self-management among elderly stroke survivors in China: a cross-sectional study” presented results of a cross-sectional study investigating four hypotheses regarding self-efficacy, self-management, and decision-making.
The study addresses a crucial problem of patients’ independence and its role in the post-stroke rehabilitation and treatment process.
The study was performed on a group of post-stroke patients hospitalized at a tertiary hospital in China. The results were reported according to Strengthening the Reporting of Observational Studies in Epidemiology (STROBE).
The subject of the research is undoubtedly interesting. Self-management and self-efficacy in elderly patients seem to be overlooked by many researchers and clinicians. At the same time, it has been proven to be significantly associated with patients’ compliance and motivation for rehabilitation (1). It was reported, that the high level of self-efficacy enhances the recovery process by incorporating more effective solutions and strategies. Interventions increasing self-efficacy should therefore be included in the post-stroke rehabilitation programs. (2)
The study protocol presented in the article is properly designed. Sample size estimation was performed according to the number of independent variables. The study included a pre-testing phase to check the questionnaires for any ambiguities needing clarification.
The article is for the most part well-composed. Methods are comprehensively described. The authors addressed the limitations of the study properly. Discussion is interesting, however requires some improvements. My main remark is, that part of the introduction (line 83 down) seems more eligible for discussion. Reconstructing these two sections would make the whole article more clear.
1. Lo SHS, Chau JPC, Lam SKY, et al. Association between participation self-efficacy and participation in stroke survivors. BMC Neurol. 2022;22(1):361. Published 2022 Sep 22. doi:10.1186/s12883-022-02883-z
2. Szczepańska-Gieracha J, Mazurek J. The Role of Self-Efficacy in the Recovery Process of Stroke Survivors. Psychol Res Behav Manag. 2020 Nov 4;13:897-906. doi: 10.2147/PRBM.S273009. PMID: 33177896; PMCID: PMC7649225.
Comments on the Quality of English LanguageRequires minor editing
